# Towards Vector Optimization on Low-Dimensional Vector Symbolic Architecture

Shijin Duan[1], Yejia Liu[2], Gaowen Liu[3], Ramana Rao Kompella[3], Shaolei Ren[2], Xiaolin Xu[1]

[1]Northeastern University, [2]University of California, Riverside, [3]Cisco Research

{duan.s, x.xu}@northeastern.edu, yliu807@ucr.edu, sren@ece.ucr.edu,
{gaoliu, rkompell}@cisco.com

Vector Symbolic Architecture (VSA) is emerging in machine learning due to its efficiency, but they are hindered by issues of hyperdimensionality and accuracy. As a promising mitigation, the Low-Dimensional Computing (LDC) method significantly reduces the vector dimension by ~100 times while maintaining accuracy, by employing a gradient-based optimization. Despite its potential, LDC optimization for VSA is still underexplored. Our investigation into vector updates underscores the importance of stable, adaptive dynamics in LDC training. We also reveal the overlooked yet critical roles of batch normalization (BN) and knowledge distillation (KD) in standard approaches. Besides the accuracy boost, BN does not add computational overhead during inference, and KD significantly enhances inference confidence. Through extensive experiments and ablation studies across multiple benchmarks, we provide a thorough evaluation of our approach and extend the interpretability of binary neural network optimization similar to LDC, previously unaddressed in BNN literature.

## 1. Introduction

Vector symbolic architecture (VSA) has been emerging for resource-limited devices, because of their low latency and high efficiency characteristics [1]. Towards practice, VSA has been applied in various applications, such as DNA coding [2], holographic feature decomposition [3], brain-computer interface tasks [4], and cognitive tasks [5]; and on different hardware including micro-controller [6], FPGA [7], and in-memory computing [8]. In VSAs, the input samples are encoded as high-dimensional vectors, i.e., $s = \text{sgn}(\sum_i \mathbf{F}_i \circ \mathbf{V}_{x_i})$, enabling parallel computations. Here, $\mathbf{F}$ and $\mathbf{V}$ are feature-related vectors. Typically, a binary VSA model needs a few megabytes, highlighting its lightweight nature. VSA has been receiving significant attention in recent years, with focuses on model design and optimization [9–12] as well as implementations in diverse fields [2, 8]. Nevertheless, further deploying binary VSA models on tiny devices with more strict resource constraints, like kilobyte-scale memory and limited computing circuits, is challenging. Since the previous training strategy is heuristic-driven, reducing vector dimensions significantly compromises the model performance. Even the basic loss functions such as cross-entropy were just explored in recent years [9, 10].

More recently, a novel VSA training strategy, low-dimensional computing (LDC) [13] has been proposed, whose vector dimension is reduced by orders of magnitude against that in previous binary VSA models (e.g., 128 vs. 10,000), without noticeable harm on the model inference accuracy. In a nutshell, LDC maps a binary VSA model to a neural network with mixed precisions (both binary and non-binary weights), thus LDC jointly trains all involved vectors in a classification task. By doing so, the trained vectors with low dimensions compress the binary VSA model from megabytes to kilobytes. Despite the overwhelming advantage, the current LDC training is still empirical and under-explored, with only basic settings, such as straight-through estimator (STE) [14]. On the other hand, while LDC is trained as a partial binary neural network (BNN), the interpretation and theoretical analysis of some critical strategies on BNN training are also absent. For example, while batch normalization (BN) and knowledge distillation (KD), as two necessities to be discussed in our work, have been implicitly employed in BNN training [14, 15], detailed analysis is still lack on them.

Second Conference on Parsimony and Learning (CPAL 2025).

In this paper, we take LDC training as an example, to analyze the training of binary VSA models, specifically for the feature vectors $\mathbf{F}$ and class vectors $\mathbf{C}$, from the gradient-based perspective. We explore the expected behavior of these vectors for optimal sample representation and classification. We also highlight how BN and KD enhance their optimization, which is neglected in standard LDC training. Besides the interpretation, we further propose two novel views: ① For BN, the floating-point operation together with trained parameters can be absorbed as channel-wise thresholds during inference, thus eliminating costly computation. ② For KD, we indicate that the temperature hyperparameter can adjust the network prediction confidence, which could also be considered during hyperparameter optimization. This can also save the extra post-processing scheme for confidence calibration that has been proposed [16].

We summarize our contributions as follows:

- We investigate the binary vector training on VSA models under the LDC training strategy, and indicate the significant benefit of batch normalization and knowledge distillation. We further demonstrate corresponding numerical validation to support our analysis.

- We depict that BN will not burden the computing during VSA inference and KD design can provide a metric to adjust the VSA prediction confidence. Not like previous conducted work, we do not require extra mechanism to achieve these goals.

- Our evaluations show that the BN and KD-assisted LDC training can achieve the best or comparable accuracies over SOTA binary VSA works, while only consuming about 2% memory footprint and $< 30\%$ latency of binary VSA models. Other ablation studies (in Appendix) are also provided to evaluate our analysis in depth.

## 2. Preliminary on Vector Symbolic Architecture

Vector symbolic architecture (VSA) represents objects using vectors and performs computation element-wise [1]. Its binary format [17] is highly efficient in computation, favoring resource-limited devices. Assuming a sample $x$ has $N$ features and each feature has $M$ discretized values, binary VSA generates the feature/value vector set $\mathbf{F} \in \{1, -1\}^{N \times D}$ and $\mathbf{V} \in \{1, -1\}^{M \times D}$ to represent $N$ feature positions and $M$ values, respectively. Binary VSA encodes one sample with a binary vector $s$,

$$s = \text{sgn}\left( \sum_{i=1}^{N} \mathbf{F}_i \circ \mathbf{V}_{x_i} \right) \tag{1}$$

where $x_i$ is the value for the $i$-th feature, $\circ$ is Hadamard product, and sgn() function binarizes the accumulation result. We set up $\text{sgn}(0) = 1$ as tie-breaker. Given a classification task, binary VSA training generates a class vector set $\mathbf{C} \in \{1, -1\}^{K \times D}$ to represent $K$ categories in this task. The similarity between vectors in $\mathbf{C}$ and $s$ is calculated by the dot product, reduced from *cosine* similarity,

$$\text{label} = \arg\max_k \mathbf{C}_k^T \, s \tag{2}$$

where the most similar one (with the highest product) is the predicted label.

Current binary VSA utilizes very high-dimensional vectors ($D \approx 10,000$) for acceptable accuracy, as $\mathbf{F}$ and $\mathbf{V}$ are generated randomly in advance [9, 10, 12]. The recently proposed strategy for binary VSA training, low-dimensional computing (LDC) [13], addresses the high-dimension issue in binary VSA by approximating the value vector mapping as a shallow neural network, $\mathcal{V}(x_i) : x_i \mapsto \mathbf{V}_{x_i}$. Then, the encoding (Eq.1) and similarity measurement (Eq.2) are expressed as a two-layer binary neural network (BNN),

$$[x_1, ..., x_N] \rightarrow \mathcal{V}(\cdot) \xrightarrow{[\mathcal{V}(x_i) \in \{1, -1\}^D]} \underbrace{y = \sum_{i=1}^{N} \mathbf{F}_i \circ \mathcal{V}(x_i)}_{\text{encoding layer}} \xrightarrow{s=\text{sgn}(y)} \underbrace{z = \mathbf{C}s}_{\text{similarity layer}} \tag{3}$$

where $\mathbf{F}$ and $\mathbf{C}$ are treated as binary weights of each layer, and sgn() as the activation between layers. Therefore, the encoding and the similarity measurement are translated to a binary weighted sum

layer and a binary linear layer, respectively. All the involved vectors, including $\mathbf{V}$(concluded from $\mathcal{V}(\cdot)$), $\mathbf{F}$, and $\mathbf{C}$ can be optimized by training this BNN rather than randomly generated. As a result, the VSA with vectors generated by LDC can achieve comparable accuracy as SOTA binary VSA [10] while only with less than $1\%$ model size, leading to extremely lightweight hardware implementation.

While LDC provides a promising solution to binary VSA training, the training principles on it are still under-explored. Previous LDC only applied the basic STE [14] and Adam [18] strategies that are used in BNN training. Besides, BNN optimization is challenging and ongoing research. Motivated by these considerations, we aim to explore current LDC training as a representative case study for BNN training, and discuss our perspective on its optimization.

Note that optimization of $\mathcal{V}(\cdot)$ is excluded from the discussion in this paper since it is a real-valued network that can be well-trained with modern strategies, and the architecture difference of $\mathcal{V}(\cdot)$ showed insignificant impact on LDC training [13]. Still, we provide an introduction to $\mathcal{V}(\cdot)$ in Appendix A. Besides, the Quantization-Aware Training (QAT) algorithm [19] is applied, iteratively freezing oscillating weights for a smooth convergence during LDC training, see Appendix B.

## 3. Training Optimization on Feature Vectors

Similar to BNN training, LDC also utilizes real-valued (or *latent*) weights for the gradient propagation, and the binarized counterparts for the forward pass. For the feature vectors $\mathbf{F}$, we denote $\mathbf{F}^r$ as the real-valued counterparts for $\mathbf{F}$ during the LDC training, which has the following property,

$$\mathbf{F}_{i,d} = \alpha(\mathbf{F}_d)\,\mathrm{sgn}(\mathbf{F}_{i,d}^r) = \begin{cases} +\alpha(\mathbf{F}_d) & \text{if } \mathbf{F}_{i,d}^r \geq 0 \\ -\alpha(\mathbf{F}_d) & \text{otherwise} \end{cases}, \text{ where } \alpha(\mathbf{F}_d) = \frac{\left\|\mathbf{F}_{:,d}^r\right\|_{l1}}{N}. \quad (4)$$

Here $\mathbf{F}_{i,d}$ means the binary element located at the $i$-th row and $d$-th column. Rather than directly applying sgn() function for binarization, a scaling factor $\alpha$ is multiplied for weight updating, which is the $l_1$-norm mean of corresponding weights. Scaling factors can be directly removed during inference. Moreover, due to the zero gradient of sgn(), it is also approximated during backward propagation,

$$(\textbf{forward})\ \mathrm{sgn}(x) = \begin{cases} 1, & x \geq 0 \\ -1, & x < 0 \end{cases}, (\textbf{backward})\ \mathrm{sgn}(x) = \begin{cases} 1, & x \geq 1 \\ x, & -1 \leq x < 1 \\ -1, & x < -1 \end{cases} \quad (5)$$

### 3.1. Analysis of Vanilla Training Process

We first explore the updating step on $\mathbf{F}$ from Eq.4 and Eq.5 in the basic LDC training. The gradient on a certain *latent* weight $\mathbf{F}_{i,d}^r$, w.r.t. the sample vector $\boldsymbol{s}$, is

$$\frac{\partial \boldsymbol{s}}{\partial \mathbf{F}_{i,d}^r} = \mathcal{V}(\boldsymbol{x}_i)_d \cdot [1(|\boldsymbol{y}_d| \leq 1) + 0(|\boldsymbol{y}_d| > 1)], \text{ where } \boldsymbol{y}_d = \sum_{i=1}^{N} \mathbf{F}_{i,d} \circ \mathcal{V}(\boldsymbol{x}_i)_d \quad (6)$$

The gradient of each weight element $\mathbf{F}_{i,d}^r$ is just based on the corresponding value vector bit $\mathcal{V}(\boldsymbol{x}_i)_d$, i.e., the $d$-th bit of the $i$-th value vector, which is $\pm 1$. Following the chain rule, the magnitude of $\mathbf{F}_{i,d}^r$ gradients can only be adjusted by the gradient from the loss term, $\partial\mathcal{L}/\partial \boldsymbol{s}$. Explicitly expressing, $|\partial\mathcal{L}/\partial\mathbf{F}_{i,d}^r| = |\partial\mathcal{L}/\partial \boldsymbol{s}_d| \cdot |\partial \boldsymbol{s}_d/\partial\mathbf{F}_{i,d}^r| = \{-|\partial\mathcal{L}/\partial \boldsymbol{s}_d|, 0, +|\partial\mathcal{L}/\partial \boldsymbol{s}_d|\}$. This is inflexible because: ① If $|\partial\mathcal{L}/\partial\mathbf{F}_{i,d}^r| \ll |\mathbf{F}_{i,d}^r|$, the $\mathbf{F}_{i,d}$ might not change by learning from the $\mathbf{F}_{i,d}^r$ updates; ② If $|\partial\mathcal{L}/\partial\mathbf{F}_{i,d}^r| \gg |\mathbf{F}_{i,d}^r|$, oscillation could happen [18] since the sign of $\mathbf{F}_{i,d}^r$ might change in every updating, hindering the training convergence.

For numerical investigation, we run 10 epochs for the LDC training, and then feed in one misclassified image (supposed to induce noticeable weight gradients) for gradient calculation. We illustrate the distributions of $\boldsymbol{y}$, $\partial\mathcal{L}/\partial\mathbf{F}^r$, and $\mathbf{F}^r$ in Figure 1(a)(b)(c), respectively. Firstly, the zero gradient makes up the most part (78.13%) of the distribution, indicating that most $\boldsymbol{y}$ fall outside the slope range $[-1, 1]$. This can be observed from Figure 1(a) as well, leading to inactively updating on *latent* weights. Then, the non-zero gradient magnitudes (as large as 0.8) are rather larger than the *latent* weights $\mathbf{F}^r$ (mostly in the range $[-0.3, 0.3]$ as shown in Figure 1(c)), which is prone to change the sign of $\mathbf{F}^r$, causing oscillation. Consequently, the $\mathbf{F}$ optimization under vanilla training is insensitive to the back-propagation and locally unstable due to large non-zero gradients.

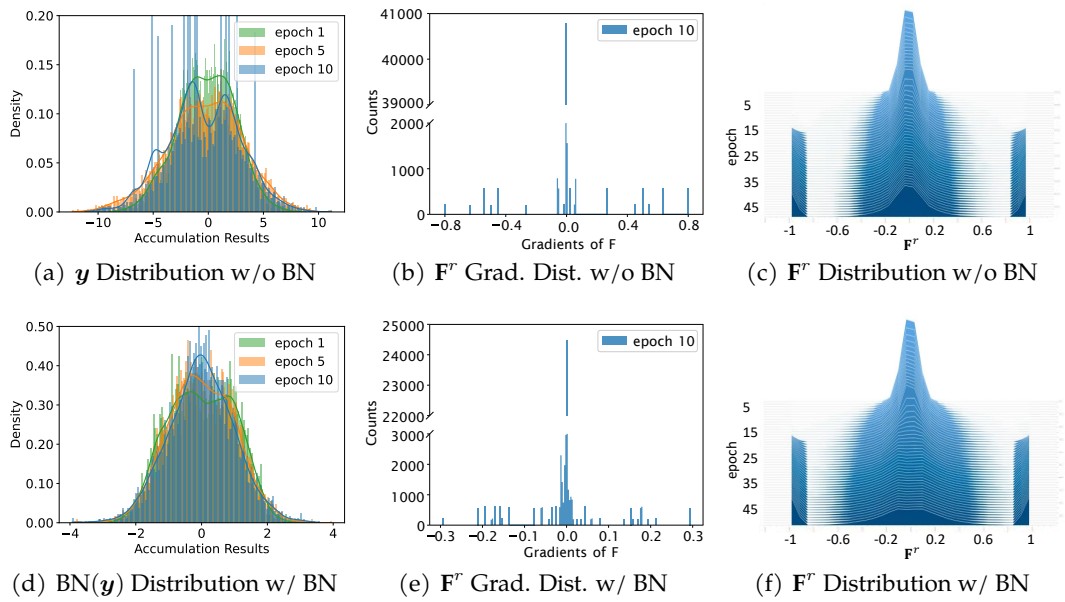

(a) $\boldsymbol{y}$ Distribution w/o BN      (b) $\mathbf{F}^r$ Grad. Dist. w/o BN      (c) $\mathbf{F}^r$ Distribution w/o BN

(d) BN($\boldsymbol{y}$) Distribution w/ BN      (e) $\mathbf{F}^r$ Grad. Dist. w/ BN      (f) $\mathbf{F}^r$ Distribution w/ BN

Figure 1: Feature vector training analysis for vanilla LDC (a)(b)(c) and our BN-based method (d)(e)(f). All figures are histograms of distributions. We run 10 epochs for (a)(b)(d)(e) to demonstrate the efficacy of BN on LDC training: BN shapes the accumulation $\boldsymbol{y}$ to zero mean and unit variance, providing more active and moderate gradients for weight updating. We run 50 epochs for (c) and (f) to show $\mathbf{F}^r$ distributino during training. The case study is tested on the LDC model with dimension $D = 64$ and on FashionMNIST.

Intuitively, we propose two simple tricks to mitigate the aforementioned issues. We directly reduce the scaling factor of $\mathbf{F}$ so that more $\boldsymbol{y}$ elements can fall into the active range $[-1, 1]$, i.e., more $\mathbf{F}^r$ with nonzero gradients. Alternatively, we can also enlarge the active range to $[-\delta, \delta]$, where $\delta > 1$, to activate more $\mathbf{F}^r$. We show the preliminary result in Table 1, with a bold configuration that $\alpha(\mathbf{F})$ is scaled down by half or enlarging the activate range to $\delta = 1.2$.

Table 1: Preliminary result on methods to mitigate the $\mathbf{F}$ training deficiencies. $\alpha(\mathbf{F}) \downarrow$ is to reduce the scaling factor. $\delta \uparrow$ is to increase the active range of sgn(). BN is the batch normalization method that we adopt in our work. We show the variance of $\boldsymbol{y}$ distribution and the range of $\partial \mathcal{L}/\partial \mathbf{F}^r$ as metrics since they directly dominate the $\mathbf{F}$ updating.

|  | LDC | $+(\alpha(\mathbf{F}) \downarrow)$ | $+(\delta \uparrow)$ | +BN |
|---|---|---|---|---|
| Var($\boldsymbol{y}$) | 10.49 | 6.83 | 10.11 | 0.98 |
| $\partial \mathcal{L}/\partial \mathbf{F}^r$ | $[-0.8, 0.8]$ | $[-0.6, 0.6]$ | $[-0.45, 0.45]$ | $[-0.3, 0.3]$ |

Both lower scaling factor and larger active range are beneficial to feature vector optimization, with lower $\boldsymbol{y}$ magnitudes (i.e., lower Var($\boldsymbol{y}$)) or smaller gradient magnitudes $|\partial \mathcal{L}/\partial \mathbf{F}^r|$. Nevertheless, these intuitive tricks are still not sufficient. The scaling of $\alpha(\mathbf{F})$ and $\delta$ should be carefully tuned as hyperparameters, which is time-consuming and tedious. On the other hand, while more active *latent* weights participate in the training, the gradient magnitude is not directly tuned by these tricks. Therefore, we emphasize the necessity of an adaptive method to solve the issues above, which is batch normalization (BN) in the feature vector training.

## 3.2. Batch Normalization Benefits Feature Vector Training

BN [20] aims to stabilize the training procedure by normalizing the activation in dimension-wise. Specifically, by applying BN to the encoding layer, we have

$$\boldsymbol{s} = \text{sgn}(\text{BN}(\boldsymbol{y})), \text{ where } \boldsymbol{y} = \sum_{i=1}^{N} \mathbf{F}_i \circ \mathcal{V}(\boldsymbol{x}_i) \text{ and } \text{BN}(\boldsymbol{y}) = \frac{\boldsymbol{y} - \text{E}(\boldsymbol{y})}{\sqrt{\text{Var}(\boldsymbol{y}) + \epsilon}} \times w_{BN} + b_{BN}. \quad (7)$$

$E(\boldsymbol{y})$ and $Var(\boldsymbol{y})$ are the statistical mean and variance of $\boldsymbol{y}$ during training, and $b_{BN}$ and $w_{BN}$ are the trainable parameters to further adjust the distribution. $\epsilon$ is a small constant for numerical stability.

With BN, the gradient on *latent* weight $\mathbf{F}^r_{i,d}$, w.r.t. the sample vector $\boldsymbol{s}$, can be derived as

$$\frac{\partial \boldsymbol{s}}{\partial \mathbf{F}^r_{i,d}} = \mathcal{V}(\boldsymbol{x}_i)_d \cdot \left(w_{BN,d}/\sqrt{Var(\boldsymbol{y}_d) + \epsilon}\right) \cdot [1(|BN(\boldsymbol{y}_d)| \leq 1) + 0(|BN(\boldsymbol{y}_d)| > 1)] \qquad (8)$$

Here, the BN contributes the gradient calculation of $\mathbf{F}^r$ in two ways: (i) appropriate affine transformation $|BN(\boldsymbol{y}_d)|$ and (ii) trainable gradient magnitude $w_{BN}/\sqrt{Var(\boldsymbol{y}_d) + \epsilon}$.

**Appropriate affine transformation** is a straightforward benefit that BN provides. Since the input of sgn() tends to approach a lower variance with BN (0.98 vs. 10.49 at epoch 10 in Figure 1(a) and (d)), more accumulations $\boldsymbol{y}$ after BN fall into the active range $[-1, 1]$, allowing more *latent* weights to participate in the updating. In Figure 1(e), only 40.63% gradients of $\mathbf{F}^r$ are still zero, which is much less than 78.13% of the case without BN in Figure 1(b). Note that $b_{BN}$ in BN is supposed to shift the accumulation $\boldsymbol{y}$, but its influence is actually negligible. This is because binary VSA exhibits centrosymmetric properties with respect to zero, i.e., vectors are from $\{-1, +1\}$ and activation function is also symmetric w.r.t. zero. Therefore, zero-centered distribution is favored for the optimization; in fact, regardless of the absence or presence of BN, the $\boldsymbol{y}$ distribution is near-zero centered, i.e., $-0.4581$ vs. $-0.0038$ in Figure 1(a) and (d).

**Trainable gradient magnitude** benefits from the trainable BN weights $w_{BN}$. Since the derivation of an optimal $w_{BN}$ is rather complicated and the space is limited, we provide a detailed analysis in Appendix C and just include a brief interpretation here. $w_{BN}$ is introduced to rescale the variance of $\boldsymbol{y}$ together with $Var(\boldsymbol{y})$, for potentially better distribution [21]. Regarding the similarity layer in LDC, the optimal input distribution can be learned by calculating $\partial \mathcal{L}/\partial BN(\boldsymbol{y})$. This leads to the optimization of BN parameter, $\partial \mathcal{L}/\partial w_{BN}$, which is under the influence of current loss and the *latent* weight $\mathbf{C}^r$. Thus, the BN weights $w_{BN}$ can adaptively scale the weight gradients on $\mathbf{F}^r$ in an appropriate range. For example in Figure 1(e), the gradients are scaled from $[-0.8, 0.8]$ (in Figure 1(b)) to $[-0.3, 0.3]$. Proper gradient magnitudes can effectively prevent weight oscillation, since the sign of *latent* weight is not prone to flip in one updating step.

As a straight comparison, we demonstrate the weight distribution of $\mathbf{F}^r$ in Figure 1(c) and (f). Throughout the training, the *latent* weights of $\mathbf{F}^r$ ~~run fast to move away from zero~~ quickly diverge from zero or are frozen to the steady state $\pm 1$ (by QAT algorithm) with the help of BN. As discussed in [19], less near-zero *latent* weights will mitigate the oscillation during BNN training. On the other hand, without BN, a big part of $\mathbf{F}^r$ stays near zero at the end of the training, indicating that the LDC training is not well converged and significant oscillation still exists in the encoding layer of LDC.

### 3.3. Batch Normalization as A Threshold

We indicate that all BN parameters can be absorbed into binarization as a threshold during binary VSA implementation. According to Eq. 1, $\boldsymbol{y} = \sum_{i=1}^{N} \mathbf{F}_i \circ \mathcal{V}(\boldsymbol{x}_i)$ is calculated for the binarization, i.e., 0 as the threshold of sgn() function. Similarly, the threshold in Eq. 7 can be expressed as

$$BN(\boldsymbol{y}) \geq 0 \Leftrightarrow \left(\sum_{i=1}^{N} \mathbf{F}_i \circ \mathcal{V}(\boldsymbol{x}_i)\right) \geq \theta, \text{ where } \theta = \left\lceil \left(E(\boldsymbol{y}) - \frac{\sqrt{Var(\boldsymbol{y}) + \epsilon} \cdot b_{BN}}{w_{BN}}\right)/\alpha_{\mathbf{F}} \right\rceil \qquad (9)$$

Therefore, the binarization operation with BN during inference can be translated to a comparison with dimension-wise thresholds $\theta$. Since the accumulation $\boldsymbol{y}_d$ on binary vectors is an integer, the threshold $\theta$ is also rounded.

**Extending to BNN.** This derivation can be generalized to the BN in BNNs as well. Hence, BN will not introduce additional computation in BNN if followed by sgn() activation, while improving the performance during inference. Prior work has developed efficient alternatives to BN [22, 23], addressing its floating-point computations during inference; yet, our analysis suggests that BN can be seamlessly integrated as dimension-wise thresholds in binary VSA, without introducing additional computational overhead.

Table 2: Top-1 accuracy of LDC training with and without normalization, with standard deviation. We consider the proposed batch normalization (BN) and other layer normalization strategies (Layer-Norm [24] and RMSNorm [25]). We vary the vector dimension $D$ of binary VSA model. The results are on 5 runs, and the best is marked as **bold**.

| Acc.(%) | $D = 32$ | 64 | 128 | 256 | 512 | 1024 |
|---------|----------|-----|------|-----|-----|------|
| LDC | $81.20^{\pm0.34}$ | $83.62^{\pm0.21}$ | $85.49^{\pm0.30}$ | $86.66^{\pm0.25}$ | $86.94^{\pm0.85}$ | $86.97^{\pm0.82}$ |
| +LayerNorm | $83.04^{\pm0.25}$ | $84.35^{\pm0.69}$ | $85.86^{\pm0.26}$ | $87.01^{\pm0.30}$ | $87.75^{\pm0.19}$ | $87.31^{\pm0.98}$ |
| +RMSNorm | $82.99^{\pm1.17}$ | $84.69^{\pm0.37}$ | $85.05^{\pm0.58}$ | $86.14^{\pm0.37}$ | $86.74^{\pm0.20}$ | $85.86^{\pm0.72}$ |
| **+BN** | $\mathbf{84.24^{\pm0.30}}$ | $\mathbf{85.52^{\pm0.37}}$ | $\mathbf{86.58^{\pm0.10}}$ | $\mathbf{87.41^{\pm0.28}}$ | $\mathbf{88.01^{\pm0.16}}$ | $\mathbf{88.53^{\pm0.17}}$ |

### 3.4. Early Evaluation

For a preliminary evaluation of batch normalization (BN) in LDC training, we compare its effectiveness against other normalization strategies across various vector dimensions, with results shown in Table 2 on the FashionMNIST dataset. All normalization methods improve training by stabilizing the process, as discussed in Section 3.2. (i) Comparing BN and vanilla training, BN not only can improve the inference accuracy of binary VSA models on various vector dimensions, but also mitigates the saturation issue existing in vanilla training when $D$ is large, e.g., $D = 512$ and 1024. Besides, vanilla training exhibits large oscillation (higher variance on accuracy) under large vector dimension. In contrast, BN-assisted LDC shows a much lower variance in accuracy, so less oscillation occurs during LDC training with BN. This also proves that BN can yield smooth convergence. (ii) When comparing normalization strategies, BN outperforms layer normalization (LN) due to its approach of normalizing across the data batch. In a data batch for the VSA model, each bit has roughly equal probabilities of being 1 and -1 along the batch. This primarily rescales the variance without introducing significant bias, as evidenced by the near-zero $b_{BN}$ (see Appendix G). In contrast, LN normalizes across vector dimensions for each sample. However, in low-dimensional VSA models, each vector bit contributes individually, and the 1/-1 distributions of vectors $s$ could vary significantly between samples across classes. This variability makes it more challenging to generalize a distribution along dimensions, potentially introducing non-zero and inconsistent biases across classes. This observation also aligns with the finding that LayerNorm slightly performs better than RMSNorm, as the latter excludes bias, which may limit its effectiveness in such scenarios. Furthermore, LN also leads to overfitting at large dimensions, as shown by increased variance and reduced accuracy from $D = 512$ to $D = 1024$, an issue BN avoids by averaging vectors in batches instead of individual dimensions. In addition, we evaluate the batch normalization performance by varying the batch size of data in Appendix F.1.

## 4. Training Optimization on Class Vectors

For the class vectors $\mathbf{C}$, we denote its *latent* weights $\mathbf{C}^r$, i.e., the real-valued counterparts, as follows:

$$\mathbf{C}_{k,d} = \alpha(\mathbf{C}) \, \text{sgn}(\mathbf{C}_{k,d}^r) = \begin{cases} +\alpha(\mathbf{C}) & \text{if } \mathbf{C}_{k,d}^r \geq 0 \\ -\alpha(\mathbf{C}) & \text{otherwise} \end{cases}, \text{ where } \alpha(\mathbf{C}) = \frac{\|\mathbf{C}^r\|_{l1}}{K \cdot D} \tag{10}$$

Here $\mathbf{C}_{k,d}$ is the binary element located at the $k$-the row and $d$-th column. Given $K$ categories and $D$ vector dimension, $\alpha(\mathbf{C})$ is the unique scaling factor for the entire class vectors.

Unlike feature vectors, the optimization on class vectors is more straightforward, since the gradient of $\mathbf{C}^r$ is directly reflected by the loss, $\partial \mathcal{L}_{CE}/\partial \mathbf{C}_{k,d}^r = -s_d(t_k - \sigma(z)_k)$, where $\sigma(z)$ is soft-max and cross-entropy is utilized in the vanilla LDC training. $t_k$ denotes the targeted probability on the $k$-th class, which is a hard target (0 or 1) in one-hot encoding. Since $s$ and $\sigma(z)$ are determined by the data input and current model weights (i.e., forward-pass information), the training optimization lies in how the loss is calculated and how the target probability $t$ is expressed.

Binary VSA models have limited capacity [26] in capturing relationships between input-target pairs due to their constrained parameter space compared to real-valued models, i.e., $2^{|\boldsymbol{\theta}|}$ vs. $\mathbb{R}^{|\boldsymbol{\theta}|}$. Consequently, the training procedure is expected to adaptively locate a more generalizable fit on

complex tasks, by prioritizing success on simpler samples and selectively forgoing those that are excessively challenging. We highlight that knowledge distillation (KD) [27] can be an ideal candidate. Specifically, KD requires an advanced network (namely teacher network) pre-trained for the current classification task, and uses its logits $\boldsymbol{z}_t$ as the criterion. The gradient of $\mathbf{C}^r$ can be calculated as

$$\frac{\partial \mathcal{L}_{KLDiv}}{\partial \mathbf{C}^r_{k,d}} = -\boldsymbol{s}_d(\sigma(\boldsymbol{z}_t/T)_k - \sigma(\boldsymbol{z}/T)_k) \cdot T \tag{11}$$

where $T$ is the temperature hyperparameter. While previous work usually takes KD as regularization during training, i.e., formulate the final loss as $\mathcal{L} = \gamma \mathcal{L}_{CE} + (1 - \gamma)\mathcal{L}_{KLDiv}$, we only consider the $\mathcal{L}_{KLDiv}$ to emphasize KD's influence in the following discussion.

## 4.1. KD Provides More Adaptive Training

Compared with the hard labels that perfectly reflect the input's true category, teacher networks might produce wrong predictions on a small number of samples. These imperfect labels actually can provide a smoother classification boundary, by recognizing the hard-to-classified samples beforehand, and replace the true labels with wrong-but-smooth probability distributions. This will mitigate the gradient magnitude of *latent* weights when LDC encounters these samples, thus reducing unnecessary weight oscillations. By "unnecessary", we interpret it as there is no need to force LDC, to learn the probability distribution that even the teacher network cannot well fit. We validate this advantage in Figure 2, where 1000 misclassified samples are selected by binary VSA after 10-epoch training, to contain some hard-to-classified samples. KD provides more near-zero gradients, mitigating the oscillation in updating $\mathbf{C}^r$ from these samples; specifically, only 2.19% of $\mathbf{C}^r$ flipped signs after updated from KLDiv loss, while 15.16% $\mathbf{C}^r$ flipped signs from CE loss. The near-zero gradients in KD are different from the case we discussed in BN, because there is no inactive range on $\mathbf{C}^r$ updating, and near-zero gradients are only caused by small losses.

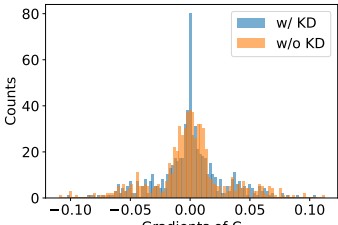

Figure 2: The $\mathbf{C}^r$ gradient distribution of binary VSA after training for 10 epochs with and without KD.

Another common benefit of KD is that it can yield a soft probability distribution, facilitating easier convergence. However, this benefit is not obvious in LDC training. We provide a quick exploration in Table 3 that applies label smoothing [28], which can just soften probability distribution, under various scaling factors. The results

Table 3: Investigating the influence of label smoothing and teacher on LDC training. $f$ is the scaling factor for label smoothing, i.e., $f\boldsymbol{t} + (1 - f)/K$. "HL-T" means hard-label from teacher.

|  | LDC | $+f = 0.1$ | $+f = 0.2$ | $+f = 0.5$ | $+$HL-T |
|---|---|---|---|---|---|
| Acc. (%) | 85.09 | 84.07 | 83.53 | 82.43 | 85.89 |

indicate that label smoothing does not positively impact the LDC training, whereas the teacher network improves performance, even with hard labels (HL-T). This disparity is likely attributed to the limited capacity of binary VSA models. Unlike current deep learning models which employ label smoothing for better generalization, binary VSA models, due to their low capacity, focus more on boundary smoothness rather than addressing overfitting issues.

## 4.2. KD Can Reshape the Confidence Distribution

From Eq.11, the temperature hyperparameter $T$ will rescale the logits $\boldsymbol{z}_t$ and $\boldsymbol{z}$, in a softened scale when $T > 1$. This introduces a previously overlooked efficacy of KD that when $T$ is large, the VSA model will have higher confidence during inference. We employ Shannon entropy [29] $H(\sigma(\boldsymbol{z})) = -\sum_k \sigma(\boldsymbol{z})_k log(\sigma(\boldsymbol{z})_k)$ to estimate the confidence for one inference. Specifically, the one-hot distribution (very confident prediction) induces the lowest entropy $H = 0$, while the uniform distribution (all $1/K$) gives the highest entropy $H = log(K)$. For a correct prediction, we expect a low entropy; on the contrary, the wrong prediction should have a high entropy so that VSA can report an unreliable prediction. We calculate the average entropy of correct predictions $\overline{H}(\sigma(\boldsymbol{z}))_T$ ($T$ for True) and wrong predictions $\overline{H}(\sigma(\boldsymbol{z}))_F$ ($F$ for False) in Table 4. In the common range of

Table 4: The confidence and accuracy of the binary VSA model trained by LDC without KD or with KD under various temperatures.

| | w/o KD | T=0.5 | T=2 | T=4 | T=8 | T=10 | T=20 |
|---|---|---|---|---|---|---|---|
| $\overline{H}(\sigma(\boldsymbol{z}))_T$ | 0.1689 | 0.4681 | 0.0654 | 0.0372 | 0.0268 | 0.0249 | 0.0263 |
| $\overline{H}(\sigma(\boldsymbol{z}))_F$ | 0.6209 | 1.0738 | 0.3883 | 0.2506 | 0.1976 | 0.1897 | 0.1879 |
| Acc. (%) | 85.09 | 84.43 | 86.17 | 86.30 | 86.51 | 86.39 | 85.07 |

$1 \leq T \leq 10$, KD shows a positive influence to produce a high-confident VSA model for correct predictions. However, the entropy of wrong predictions also decreases along $T$, which is against our expectation for an ideal model. Nevertheless, the correct prediction can always show a lower entropy (meaning high confidence) than the wrong predictions on all $T$ selections. Note that the entropy of predictions under $T = 0.5$ is larger than the case without KD, because $T < 1$ temperature will sharpen the probability distribution. Consequently, we propose another metric to evaluate the performance of KD, i.e., the prediction confidence that a VSA model can provide. This can be jointly considered with accuracy as a trade-off when designing the temperature $T$. To evaluate distillation in a more general context, we further explore teacher networks with different capacities in Appendix F.2, and different divergence losses in LDC training in Appendix F.3.

## 5. Evaluation

**Datasets.** We select representative datasets for binary VSA models. ISOLET [30] and HAR [31] are two commonly evaluated benchmarks in previous VSA work, where ISOLET is a voice recording collection and HAR is an activity gesture collection. Besides, we evaluate two other lightweight applications, i.e., seizure detection (CHB-MIT) [32] on brain-computer interface and credit card fraud detection (CreditCard) [33] for portable devices. Also, we include FashionMNIST [34] since it is the most challenging one in previous VSA work. A detailed description is provided in Appendix D.

**Training Setup.** As basic configuration, we follow the LDC training setup [13]. We apply batch normalization and knowledge distillation with their default setup. For KD, we still set $\gamma = 0$ (without considering $\mathcal{L}_{CE}$) to emphasize the KD benefit and $T = 4$ as a common choice. We employ 3-layer MLP on the first four datasets (with accuracies in order 96.54%, 97.08%, 99.10%, and 94.90%, respectively), since they are all 1-D signal samples; and use ResNet-18 for the KD on FashionMNIST (with accuracy 92.51%). Other teacher networks are evaluated in Appendix E.

**Model Comparison.** We compare our strategy (noted as "LDC+BNKD") with the vanilla LDC, and other SOTA binary VSA works [9, 10, 12]. For LDC+BNKD and LDC, we evaluate them on different vector dimensions $D$, while we keep the configuration of $\mathcal{V}(\cdot)$ fixed as suggested, i.e., 4 [13]. For other binary VSA models, we assume the dimension $D = 10,000$, as suggested in their works.

### 5.1. Inference Accuracy

We demonstrate the inference accuracy of related binary VSA works in Table 5. The original LDC training generally has slightly worse inference performance than SOTA binary VSA works. Also, the saturation issue is also apparent on LDC, i.e., little accuracy increment by doubling the vector dimension from 256 to 512. This highlights the necessity of investigation on better VSA training strategy, rather than directly increasing vector dimensions. On the other hand, the LDC training with BN and KD assistance can obviously improve inference performance. While BN and KD can mitigate but not entirely eliminate the saturation issue on certain tasks (e.g., FashionMNIST), we advocate for architectural enhancements to binary VSA models to fundamentally augment their capabilities. In the breadth view, "LDC+BNKD" demonstrates superior accuracy on two benchmarks while maintaining a performance gap of $< 1\%$ compared to the highest-performing SOTA VSA models on two other benchmarks. Therefore, the BNKD-assisted LDC training can provide an LDC model with

Table 5: Inference accuracy comparison between SOTA VSA works, LDC, and our training strategy. We assume $D = 10,000$ for binary VSA models (the first four), and vary the dimension $D = (64, 256, 512)$ for LDC and our method. Best results are in **bold**, and second bests are underlined.

| Model Accuracy (%) | QuantHD [12] | G($2^3$)-VSA[9] | G($2^4$)-VSA[9] | LeHDC [10] | LDC[13] 64 | LDC[13] 256 | LDC[13] 512 | LDC+BNKD 64 | LDC+BNKD 256 | LDC+BNKD 512 |
|---|---|---|---|---|---|---|---|---|---|---|
| ISOLET | 92.70 | 94.40 | **96.00** | 94.89 | 88.26 | 92.97 | 93.70 | $88.72^{\pm0.46}$ | $93.87^{\pm0.34}$ | $94.28^{\pm0.28}$ |
| HAR | 91.25 | 95.60 | **96.60** | 95.23 | 93.08 | 94.67 | 94.90 | $93.66^{\pm0.54}$ | $95.20^{\pm0.30}$ | $95.64^{\pm0.25}$ |
| CHB-MIT | 86.57 | N/A | N/A | 91.16 | 95.30 | 96.39 | 96.53 | $97.32^{\pm0.65}$ | $98.01^{\pm0.56}$ | $98.04^{\pm0.35}$ |
| CreditCard | **94.69** | N/A | N/A | 93.88 | 93.37 | 93.88 | 93.88 | $94.19^{\pm0.46}$ | $94.49^{\pm0.23}$ | $94.69^{\pm0.36}$ |
| FashionMNIST | 80.26 | 86.70 | 87.40 | 87.11 | 83.62 | 86.66 | 86.94 | $86.48^{\pm0.22}$ | $88.38^{\pm0.21}$ | $88.91^{\pm0.10}$ |

Table 6: The inference accuracy on datasets by varing $\gamma$, assuming the vector dimension $D = 64$ for our LDC+BNKD model. Results are averaged on 5 runs. The best results for each benchmark are marked as **bold**, and the second bests are underlined.

| Acc. (%) | $\gamma =0.0$ | 0.2 | 0.4 | 0.6 | 0.8 | 1.0 |
|---|---|---|---|---|---|---|
| ISOLET | $\mathbf{88.92}^{\pm0.36}$ | $88.02^{\pm0.41}$ | $88.08^{\pm0.91}$ | $87.66^{\pm0.97}$ | $88.06^{\pm0.52}$ | $88.30^{\pm0.81}$ |
| HAR | $93.64^{\pm0.58}$ | $93.71^{\pm0.26}$ | $93.56^{\pm0.24}$ | $\mathbf{94.01}^{\pm0.37}$ | $93.60^{\pm0.24}$ | $93.48^{\pm0.46}$ |
| CHB-MIT | $97.14^{\pm0.64}$ | $\mathbf{97.92}^{\pm0.42}$ | $97.35^{\pm0.48}$ | $97.33^{\pm0.67}$ | $97.14^{\pm0.65}$ | $96.84^{\pm0.78}$ |
| CreditFraud | $94.08^{\pm0.58}$ | $94.39^{\pm0.81}$ | $\mathbf{94.49}^{\pm0.67}$ | $93.98^{\pm0.67}$ | $94.08^{\pm0.46}$ | $92.86^{\pm0.72}$ |
| FashionMNIST | $86.30^{\pm0.25}$ | $86.37^{\pm0.13}$ | $86.34^{\pm0.26}$ | $\mathbf{86.39}^{\pm0.15}$ | $86.13^{\pm0.43}$ | $85.37^{\pm0.38}$ |

inference performance comparable to that of SOTA VSA work. Notably, this superiority is achieved with only 1/20 vector dimensions of VSA models, i.e., $D = 512$ vs. $D = 10,000$.

## 5.2. Trade-off Between $\mathcal{L}_{CE}$ and $\mathcal{L}_{KLDiv}$

To evaluate the trade-off between the knowledge from the ground-truth label and the advanced framework, we varies $\gamma$ in the final KD loss, $\mathcal{L} = \gamma\mathcal{L}_{CE} + (1 - \gamma)\mathcal{L}_{KLDiv}$. The accuracy results are given in Table 6. The differences between various $\gamma$ are relatively low for most benchmarks (but indeed significant in some such as CreditFraud and FashionMNIST). Nevertheless, the efficacy of KD is obvious on most benchmarks; without KD, i.e., $\gamma = 1$, LDC is prone to perform the worst. On the other hand, all benchmarks have the best performance when $\gamma \leq 0.6$, meaning that partly inducing $\mathcal{L}_{KLDiv}$ are likely to give an inference improvement for LDC training.

## 5.3. Hardware Preparation

Since binary VSA is well-known for its ultra-lightweight and real-time implementation. It is necessary to estimate the memory footprint and potential hardware latency. We calculate the memory requirement of involved binary vectors, including **V**, **F**, and **C**. We also evaluate the latency by calculating model's circuit-depth complexity (CDC). Its calculation on binary VSA is discussed in [9]. We take the ISOLET and FashionMNIST as two examples to evaluate the hardware performance in Table 7, and the results of other datasets are available in Appendix F.5. ① SOTA binary VSA models, e.g., QuantHD, G-VSA, and LeHDC have heavy-loaded hardware

Table 7: The memory footprint (in KB) and hardware latency (in CDC) for different binary VSA models, evaluated on ISOLET and FashionMNIST.

| | ISOLET Mem. (KB) | ISOLET CDC | FashionMNIST Mem. (KB) | FashionMNIST CDC |
|---|---|---|---|---|
| QuantHD | 1124 | 295 | 1313 | 295 |
| LeHDC | 1124 | 295 | 1313 | 295 |
| G($2^3$)-VSA | (1058)* | 402 | (998)* | 405 |
| G($2^4$)-VSA | (1410)* | 430 | (1330)* | 434 |
| LDC-64 | 5.27 | 73 | 6.48 | 74 |
| LDC-256 | 20.70 | 118 | 25.54 | 119 |
| LDC-512 | 41.28 | 145 | 50.94 | 146 |
| **LDC+BNKD-64** | **5.35** | **73** | **6.56** | **74** |
| **LDC+BNKD-256** | **21.02** | **118** | **25.86** | **119** |
| **LDC+BNKD-512** | **41.92** | **145** | **51.58** | **146** |

*Not given in original work, but estimated by us.

overhead, because they have very large vector dimensions, $D = 10,000$, showing the deficiency of current high-dimension VSA models. ② In contrast, low-dimension VSA models (LDC and LDC+BNKD) have much lower memory footprint and hardware latency. They can achieve 1/20~1/260 memory

usage compared to SOTA VSA models; yet our LDC+BNKD strategy will induce slightly larger ($1.2\% \sim 1.6\%$) memory than their LDC counterparts, because we store the thresholds derived from BN. Nevertheless, this memory increment is negligible and can be easily handled by tiny devices. Jointly, LDC+BNKD and LDC have only $17\% \sim 49\%$ hardware latency (CDC) compared to those high-dimension VSA models, depicting their real-time advantage. Correspondingly, we also demonstrate the real inference time of different binary VSA models in Appendix F.5. Further, LDC+BNKD has the same latency as its LDC counterpart, because the BN and KD training does not introduce extra computation.

We present other evaluations of KD temperature and model robustness in Appendix F.4 and F.6, and further evaluate the effectiveness of BN and KD on BNNs in Appendix F.7.

## 6. Conclusion

Low-dimensional computing (LDC) has been put forth as a promising training strategy for binary VSA, due to its ability to generate VSA with ultra low-dimensional vectors. However, the potential for LDC training remains a lack of exploration. In this paper, we offer a thorough analysis of vector optimization in LDC and highlight the indispensability of batch normalization (BN) and knowledge distillation (KD) during training. We argue that BN does not add computational overhead to hardware implementations, and that KD positively impacts confidence calibration during training. Our analysis can also be extended to general BNN, thereby helping theoretical analysis of the roles of BN and KD in BNN optimization. Further discussion is provided in Appendix G.

## Acknowledgements

Shijin Duan and Xiaolin Xu were supported in part by the U.S. NSF under Grants CNS-2326597, CNS-2239672, and a Cisco Research Award. Yejia Liu and Shaolei Ren were supported in part by the U.S. NSF under Grants CNS-2326598.

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

## A. Value Vector Mapping

In LDC, the vector mapping of an input value is expressed as a shallow neural network, namely *ValueBox*, $\mathcal{V}(\boldsymbol{x}_i) : \boldsymbol{x}_i \mapsto \mathbf{V}_{\boldsymbol{x}_i}$. The structure of $\mathcal{V}(\boldsymbol{x}_i)$ applied in our work is

$$\boldsymbol{x}_i \rightarrow \text{Linear}(1,20) \rightarrow \text{BN}(20) \xrightarrow{tanh()}$$

$$\text{Linear}(1,D_{\mathbf{v}}) \xrightarrow{sgn()} \mathbf{v} \xrightarrow{duplicate} \mathbf{V}_{\boldsymbol{x}_i}$$

Here, $D_{\mathbf{v}}$ is the output dimension of the second linear layer, which is the dimension of the value vector. As indicated in the original paper [13], the $D_{\mathbf{v}}$ can be unequal to the vector dimension $D$ of $\mathbf{F}$ and $\mathbf{C}$, but only requiring $D$ is multiples of $D_{\mathbf{v}}$. Therefore, after the network output, $\mathbf{v}$, this binary vector is duplicated by $D/D_{\mathbf{v}}$ times as the final output of *ValueBox*, $\mathbf{V}_{\boldsymbol{x}_i}$, to align with the vector dimension of $\mathbf{F}$ and $\mathbf{C}$. This trick can save the memory footprint and results from the observation that value vectors usually do not need as large dimensions as feature and class vectors. For the $N$ input features, only one unique *ValueBox* is generated. The *ValueBox* later is translated to a Look-up Table, recording the binary vector outputs for all possible input values.

**Remark: depth of $\mathcal{V}(x)$ network.** The original LDC work [13] explores several designs for the value mapping function $\mathcal{V}(x) : x \mapsto \mathbf{V}_x$ (see Fig. 6 in [13]), experimentally concluding that a shallow network is sufficient to express value mapping. While the original paper does not analyze this approximation, we offer our perspective that it may be related to the "value importance" in classification, where $\mathcal{V}(x)$ determines the optimal segmentation of values from 1 to $M$. Each segment corresponds to a single vector representation, and values within a segment have similar importance for classification. For instance, $\mathcal{V}(x)$ could divide the range $[0, 255]$ into segments such as $\{[1, 50], [51, 100], [101, 150], [151, 200], [201, 255]\}$, corresponding to vectors $\{0000, 0001, 0011, 0111, 1111\}$ for values in each segment. The optimization of segmentations is done automatically during LDC training. This mapping is relatively easy to optimize, given the characteristics of data in VSA tasks, so a shallow network can satisfy the approximation of $\mathcal{V}(x)$. Note that more complicated value distributions could require a more complex $\mathcal{V}(x)$ architecture, e.g. forcing above segments to generate vector set $\{0000, 0101, 0011, 0110, 1111\}$, deeper network may be needed to fit this projection. However, observations in [13] indicate that a shallow network is adequate for current VSA tasks, as value-vector projections are not as complicated as aforementioned. Following the original LDC design, we adopt a **two-layer** network in our work.

## B. Training Details

Except for the basic training setup shown in the paper, we provide other training information here. We apply Adam as the optimizer with 1e-3 initial learning rate, which is linearly decayed to zero during training. We run our experiments for 50 epochs to guarantee convergence. The gradient of *latent* weights are clipped within the magnitude $\pm 1$.

For the quantization-aware training (QAT) strategy [19], we briefly show the main idea here for quick reference. QAT first defined the oscillation of a certain *latent* weight $w$, in our binary training scenario, by satisfying two conditions: ① the sign of $w$ flipped between two consecutive iterations $t - 1$ and $t$, i.e., $\Delta^t = \text{sgn}(w^t) - \text{sgn}(w^{t-1}) \neq 0$. Here $\Delta^t = 2$ means the sign of $w$ is changed from -1 to 1, $\Delta^t = -2$ means the opposite, and $\Delta^t = 0$ represents no sign flipping. ② one oscillation is the sign flipping directions between two consecutive iterations are opposite, i.e., $o^t = (\Delta^t \neq \Delta^{t-1}) \cdot (\Delta^t \times \Delta^{t-1} \neq 0)$. Thus, $o^t = 1$ means there is an oscillation in iteration $t$, and 0 otherwise. The frequency of oscillations over iterations is tracked using the exponential moving average (EMA):

$$f^t = m \cdot o^t + (1 - m) \cdot f^{t-1}$$

For the iterative weight freezing on QAT, we define a threshold $f_{th}$ that once the oscillation frequency of a certain weight exceeds $f_{th}$, this weight will be frozen and not updated for the rest of training. To minimize the difference in computations between forward and backward [18], we set the frozen state of a *latent* weight $w_z$ as $w_z = \text{sgn}(w)$. Therefore, the frozen *latent* weights will be equal to their

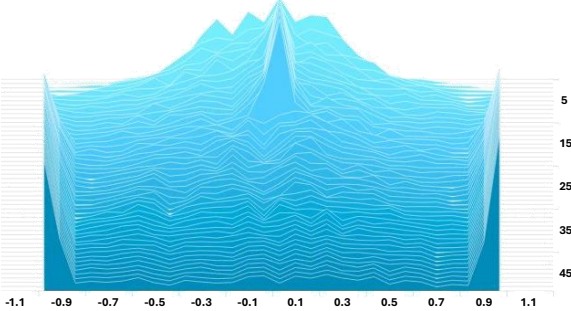

Figure 3: The $\mathbf{C}^r$ distribution along 50 epochs.

signed results. Note that the scaling factor $\alpha$ computing will exclude the frozen weights, i.e., we only calculate the $l_1$-norm mean of active *latency* weights. In our experiments, we empirically set $m = 0.01$ and $f_{th} = 0.02$. And in order to make QAT effective when LDC and LDC+BNKD tend to converge, we activate QAT at epoch 15 until the end.

**Remark: advantage of LDC training over post-training quantization (PTQ).** LDC training on VSA [13] is a QAT strategy, which generally outperforms PTQ in accuracy, albeit being more compute-intensive. However, thanks to the lightweight nature of the LDC model with a kilobyte-sized architecture, QAT training is highly efficient, taking only about 5 minutes on an NVIDIA 3070 GPU, as measured in our experiments. Regarding the similar training procedure in BNN, QAT is also widely adopted [35] for its ability to effectively handle the highly discrete nature of binarization, which is more extreme than integer quantization (e.g., 8-bit).

## C. Trainable Gradient Magnitude

We prove why the parameter $w_{BN}$ in batch normalization is optimized on both forward and backward information. Therefore, $w_{BN}$ has more adaptive adjustment on the encoding accumulations. Recall that the LDC model with BN can be expressed as, starting from the accumulation result $\boldsymbol{y}$,

$$\boldsymbol{z} = \mathbf{C} \times \text{sgn} \left( \frac{\boldsymbol{y} - \text{E}(\boldsymbol{y})}{\sqrt{\text{Var}(\boldsymbol{y}) + \epsilon}} \times w_{BN} + b_{BN} \right)$$

We still assume the LDC model is trained on softmax activation and cross-entropy loss, then after feeding in one query sample, the derivative of the $d$-th element $w_{BN,d}$ w.r.t. $\mathcal{L}_{CE}$ is

$$\frac{\partial \mathcal{L}_{CE}}{\partial w_{BN,d}} = \frac{\partial \mathcal{L}_{CE}}{\partial \boldsymbol{s}_d} \frac{\partial \boldsymbol{s}_d}{\partial w_{BN,d}}$$

$$\text{where } \frac{\partial \mathcal{L}_{CE}}{\partial \boldsymbol{s}_d} = - \left[ \sum_k \mathbf{C}_{k,d}^r (\boldsymbol{t}_k - \sigma(\boldsymbol{z})_k) \right],$$

$$\frac{\partial \boldsymbol{s}_d}{\partial w_{BN,d}} = \frac{\boldsymbol{y}_d - \text{E}(\boldsymbol{y}_d)}{\sqrt{\text{Var}(\boldsymbol{y}_d) + \epsilon}} \cdot \left[ 1(|\text{BN}(\boldsymbol{y}_d)| \le 1) + 0(|\text{BN}(\boldsymbol{y}_d)| > 1) \right].$$

Here we boldly assume the discussed optimizing surface of $w_{BN,d}$ is not constant, i.e., $|BN(\boldsymbol{y})| \le 1$; otherwise the gradient is always zero, disabling updating. The term in $\partial \mathcal{L}_{CE}/\partial w_{BN,d}$ related to $w_{BN,d}$ is $\sum_k \mathbf{C}_{k,d}^r \sigma(\boldsymbol{z})_k$; this term is not monotonous to $w_{BN,d}$ since $\mathbf{C}_{:,d}^r$ are not constant and even does not has the same sign, as shown in Figure 3. This conclusion can be also described as the weighted sum of softmax members is not monotonous. Therefore, the $w_{BN,d}$ optimization is not convex, and only the local optimum can be derived. Besides, since the non-linearity of softmax, there is no analytical solution for the equation $\partial \mathcal{L}_{CE}/\partial w_{BN,d} = 0$. Nevertheless, if numerically solving this equation, it can be equivalent to $\mathbf{C}_{l,d}^r = \sum_k \mathbf{C}_{k,d}^r \sigma(\boldsymbol{z})_k$ assuming the true label of this query sample is $l$. From this analysis, we can derive that the local optimum of $w_{BN,d}$ highly relies on the

*latent* weight $\mathbf{C}^r$. This gives a valuable perspective that the BN can be well adjusted from its followed layers through backward propagation, while the normal regularization $\mathrm{E}(\boldsymbol{y})$ and $\mathrm{Var}(\boldsymbol{y})$ can only statistically reshape the accumulation based on feedforward information.

## D.  Supplementary of Evaluation Setup

Table 8: Configurations of the evaluated datasets.

| Dataset | ISOLET | HAR | CHB-MIT | CreditCard | FashionMNIST |
|---|---|---|---|---|---|
| # of features ($N$) | 617 | 561 | 1472 | 29 | 784 |
| # of (train, test, class) | (6238, 1559, 26) | (7352, 2947, 6) | (13920, 664, 2) | (3940, 196, 2) | (60000, 10000, 10) |

We provide the basic information of our selected benchmarks in Table 8. Since LDC models (and even current high-dimension VSA models) have limited capacity, due to their straightforward and simple calculation, they are specifically proposed for those tasks that are not difficult to classify but still require rather limited hardware resourcesand real-time response. For example, the classification of segmented signals and simple images on portable devices is favorable to LDC. Therefore, we select the four representative datasets covering different lightweight scenarios. For image classification, binary VSA models so far can only handle easy image recognition with just passable accuracy, such as MNIST[36] and FashionMNIST[34], thus we did not include complex tasks, such as CIFAR[37] or ImageNet[38] in our evaluation.

## E.  Knowledge Distillation under Ensemble Models

Table 9: The inference accuracy of benchmarks by training LDC with ensemble-model KD, assuming the vector dimension $D = 64$ for LDC+BNKD and the size of ensemble is $G = 10$. Results are averaged on 5 runs.

| Acc. (%) | ISOLET | HAR | CHB-MIT | CreditFraud | FashionMNIST |
|---|---|---|---|---|---|
| $D = 64$ | $88.05^{\pm 0.49}$ | $93.51^{\pm 0.22}$ | $96.57^{\pm 0.58}$ | $92.04^{\pm 1.47}$ | $84.32^{\pm 0.20}$ |
| 256 | $93.14^{\pm 0.44}$ | $95.27^{\pm 0.20}$ | $98.77^{\pm 0.29}$ | $92.76^{\pm 0.67}$ | $87.75^{\pm 0.12}$ |
| 512 | $94.41^{\pm 0.05}$ | $95.74^{\pm 0.21}$ | $98.73^{\pm 0.17}$ | $93.37^{\pm 0.36}$ | $88.62^{\pm 0.13}$ |

There are many knowledge distillation strategies that have been discussed [39]. As a consensus, the teacher and the student networks have the similar architecture, so that the transferred knowledge can be well mimicked by the student. However, there is currently no developed model akin to the LDC. Thus, we also try to use the ensemble of several LDC models as the teacher network, so the teacher network can perform better than a single LDC model and has a similar classification boundary of which the LDC training is capable as well. As a general approach for ensemble knowledge distillation, the *average* rule is mostly applied in development. Specifically, the ensemble categorical probability we utilized for $k$-th class is

$$\sigma\left(\boldsymbol{z}_t\right)_k = \frac{1}{G} \cdot \sum_{g=1}^{G} \sigma\left(\boldsymbol{z}_t^g\right)_k \tag{12}$$

assuming there are $G$ teacher models. Since the ensemble strategy averages the prediction, the bias and noise on boundaries during training are mitigated, so that the ensemble boundary is more of generalization. We present the LDC+BNKD training results in Table 9, which includes 10 LDC models as the teachers. Compared with the result we demonstrated in Table 5 in the main paper, neither an ensemble model nor an advanced network consistently outperforms the other as the teacher model. The two strategies have similar performance on ISOLET and HAR; however, the ensemble strategy has a slight advantage on the CHB-MIT dataset while the advanced-network strategy has obvious superiority on CreditFraud and FashionMNIST datasets. Especially for FashionMNIST, low dimensional models, e.g., $D = 64$, the real-valued advanced network has a significant advantage (as large as 2% accuracy gap) over the ensemble model.

# F. Supplementary on Evaluation

## F.1. Batch Normalization Performance under Different Batch Size

Since BN normalizes across data batches, it is important to evaluate the accuracy of LDC+BN under various batch sizes, exploring whether the normalization strategy can be applied to different training configurations. Table 10 presents the results of applying LDC training with $D = 64$ on FashionMNIST. Across varying batch sizes, the accuracy remains consistent with minor fluctuations, i.e., there is no degradation on small batches. This suggests that batch size has minimal influence on normalization during LDC training. This behavior highlights the efficiency and flexibility of BN in LDC, as it maintains performance consistently across different training configurations.

Table 10: The inference accuracy of LDC+BN under different batch sizes. Results are averaged on 5 runs.

|          | 32    | 64    | 128   | 256   |
|----------|-------|-------|-------|-------|
| **LDC+BN** | 85.28 | 85.52 | 85.04 | 85.79 |

## F.2. Distillation Performance under Different Capacity Mismatch

Table 11: The inference accuracy of LDC+KD with different teacher networks, with the accuracy of each teacher network provided in parentheses.

|          | –     | MLP(89.21) | AlexNet(92.14) | ResNet18(92.51) | ResNet50(92.65) |
|----------|-------|-----------|----------------|-----------------|-----------------|
| LDC+KD   | 83.62 | 83.89     | 85.41          | 86.30           | 86.13           |

Since LDC distillation relies on learning knowledge from the teacher network, it is important to investigate how the capacity of teacher networks affects distillation performance. Table 11 presents a comparison of results from teacher networks with varying capacities using the FashionMNIST dataset, taking vector dimension $D = 64$ and distillation temperature $T = 4$ as an example. The results show that the complexity and capacity of the teacher network significantly influence LDC+KD performance. As the teacher becomes more advanced (e.g., from MLP to ResNet18), it captures task knowledge more effectively, improving the accuracy of LDC+KD. However, overly large teacher networks, such as ResNet50, provide little additional benefit compared to ResNet18, and LDC struggles to extract more useful knowledge from them. Therefore, selecting a teacher network requires balancing model capacity with training effort to optimize both efficiency and effectiveness.

## F.3. Distillation Performance under Other Divergence Loss

Table 12: The inference accuracy of the binary VSA model trained by LDC under KLDiv or JSDiv [40] loss, with various temperatures.

|       | $T = 0.5$ | 2     | 4     | 8     | 10    | 20    |
|-------|-----------|-------|-------|-------|-------|-------|
| KLDiv | 84.43     | 86.17 | 86.30 | 86.51 | 86.39 | 85.07 |
| JSDiv | 85.03     | 85.40 | 86.05 | 85.23 | 85.59 | 84.26 |

Due to the significant capacity mismatch between low-dimensional VSA models and the teacher networks, KL divergence (KLDiv) may lead to "mode-seeking" during distillation [41, 42]. To address this, we investigate whether a more generalized divergence metric, such as Jensen-Shannon divergence (JSDiv), could mitigate this issue. Table 12 compares the performance of KLDiv and JSDiv under different temperatures. Our results show that KLDiv and JSDiv perform similarly in LDC training, but when $T > 1$, KLDiv achieves slightly better results despite JSDiv being the more generalized metric. This may be because (i) VSA tasks are less complex than modern vision or language tasks, making the mode distribution simpler and allowing KLDiv to capture dominant modes effectively, which the student network can learn. (ii) Less significant modes captured by the teacher network might be ignored by the student, enabling the student to focus on dominant modes, improving performance with KLDiv. This observation aligns with our discussion in Section 4.1, where KD encourages the student to prioritize easier-to-classify samples while de-emphasizing harder ones.

## F.4. Performance of BNKD under Different Distillation Temperatures

Table 13: The inference accuracy on datasets by varying temperature T in KD, assuming the vector dimension $D = 64$ for LDC+BNKD. Results are averaged on 5 runs.

| Acc. (%) | $T = 0.5$ | 2 | 4 | 8 | 10 | 20 |
|---|---|---|---|---|---|---|
| ISOLET | $87.44^{\pm 0.45}$ | $88.34^{\pm 0.42}$ | $\mathbf{88.65}^{\pm 0.77}$ | $87.11^{\pm 0.62}$ | $86.79^{\pm 0.92}$ | $84.84^{\pm 0.62}$ |
| HAR | $93.24^{\pm 0.52}$ | $93.67^{\pm 0.29}$ | $\mathbf{93.84}^{\pm 0.27}$ | $93.46^{\pm 0.18}$ | $93.25^{\pm 0.25}$ | $93.23^{\pm 0.26}$ |
| CHB-MIT | $96.84^{\pm 0.83}$ | $97.14^{\pm 0.60}$ | $97.05^{\pm 0.64}$ | $\mathbf{97.68}^{\pm 0.81}$ | $96.93^{\pm 0.78}$ | $96.51^{\pm 0.50}$ |
| CreditFraud | $93.47^{\pm 0.91}$ | $93.78^{\pm 0.67}$ | $\mathbf{94.39}^{\pm 0.43}$ | $93.16^{\pm 0.93}$ | $93.98^{\pm 0.56}$ | $94.08^{\pm 0.46}$ |
| FashionMNIST | $84.43^{\pm 0.44}$ | $86.17^{\pm 0.30}$ | $86.30^{\pm 0.34}$ | $\mathbf{86.51}^{\pm 0.18}$ | $86.39^{\pm 0.22}$ | $85.07^{\pm 0.19}$ |

We further demonstrate the temperature influence on the LDC+BNKD model for other benchmarks, which is shown in Table 13. While temperature could have a significant impact on the accuracy (e.g., on ISOLET), empirical temperatures, such as $T = 4$, indeed can perform best in most cases. On the other hand, if the temperature falls outside the common range $T \in [1, 10]$ that is suggested in previous works, the accuracy usually is compromised a little. While current KD works commonly use $T = 4$ as a default choice, there is no theoretical analysis on temperature optimization, which could be a potential topic, especially for binary neural networks.

## F.5. Hardware Preliminaries for Other Datasets

Table 14: The memory footprint (in KB) and hardware latency (in CDC) for different binary VSA models, evaluated on HAR, CHB-MIT, and CreditCard. "ours" refers to the "LDC+BNKD".

| | HAR | | CHB-MIT | | CreditCard | |
|---|---|---|---|---|---|---|
| | Mem. (KB) | CDC | Mem. (KB) | CDC | Mem. (KB) | CDC |
| QuantHD | 1029 | 295 | 2163 | 296 | 359 | 291 |
| LeHDC | 1029 | 295 | 2163 | 296 | 359 | 291 |
| $G(2^3)$-VSA | $(983)^\star$ | 401 | $(968)^\star$ | 414 | $(968)^\star$ | 363 |
| $G(2^4)$-VSA | $(1310)^\star$ | 428 | $(1290)^\star$ | 445 | $(1290)^\star$ | 377 |
| LDC-64 | 4.66 | 73 | 11.92 | 75 | 0.38 | 69 |
| LDC-256 | 18.27 | 118 | 47.30 | 120 | 1.12 | 134 |
| LDC-512 | 36.42 | 145 | 94.46 | 147 | 2.11 | 141 |
| **ours-64** | **4.74** | **73** | **12.01** | **75** | **0.42** | **69** |
| **ours-256** | **18.59** | **118** | **47.65** | **120** | **1.28** | **134** |
| **ours-512** | **37.06** | **145** | **95.17** | **147** | **2.43** | **141** |

$^\star$Not given in original work, but estimated by us.

Besides the ISOLET and FashionMNIST, we further demonstrate the hardware overhead of other datasets in our paper, in Table 14. The hardware performance results are akin to the observation we get from the main paper. Besides, there is another interesting observation that, compared with other high-dimension VSA models (QuantHD and LeHDC), G-VSA seems to occupy more memory when the input samples have a small number of features, such as CreditCard ($N = 29$), while it has less memory footprint when the dataset has a large number of features, such as CHB-MIT ($N = 1472$). This is because G-VSA does not generate feature vector set **F** but uses permutations to encode value vectors **V**. Nevertheless, LDC and LDC+BNKD models still have overwhelming advantages on the hardware overhead against other VSA models.

Further, we exemplarily report the actual inference time of different binary VSA models on ISOLET and FashionMNIST dataset, corresponding to CDC results. The actual inference time is shown in Table 15, where the data input has a batch size of 100, with vector dimensions of 10,000 for (QuantHD, LeHDC, G-VSA) and 64 for (LDC, LDC+BNKD). Similar to the results in Table 7, low-dimensional VSA models (LDC, LDC+BNKD) significantly outperform high-dimensional VSA models. G-VSA requires more time than QuantHD and LeHDC due to its multi-bit vector computations, whereas QuantHD and LeHDC rely on binary vectors. Among the G-VSA models, $G(2^3)$-VSA and $G(2^4)$-VSA exhibit similar runtimes, despite the latter using more bits, thanks to optimized integer kernels on hardware. Additionally, GPU execution is considerably faster than CPU execution, owing to parallel processing and superior computational capabilities.

Table 15: Actual inference time comparison of different VSA models, on GPU and CPU devices. CPU tests are run on a 3.80GHz 16-core Intel i7-10700K, and GPU tests are run on an NVIDIA 3070 GPU. Time is measured in microseconds.

| | Latency ($\mu$s) | QuantHD, LeHDC | $G(2^3)$-VSA | $G(2^4)$-VSA | LDC-64, LDC+BNKD-64 | LDC-256, LDC+BNKD-256 | LDC-512, LDC+BNKD-512 |
|---|---|---|---|---|---|---|---|
| GPU | ISOLET | 43.6 | 63.8 | 63.0 | 0.18 | 0.61 | 1.36 |
| | FashionMNIST | 55.0 | 77.7 | 77.8 | 0.22 | 0.78 | 1.74 |
| CPU | ISOLET | 102.7 | 830.3 | 825.6 | 4.6 | 18.2 | 33.9 |
| | FashionMNIST | 131.3 | 975.2 | 994.6 | 5.8 | 23.7 | 44.7 |

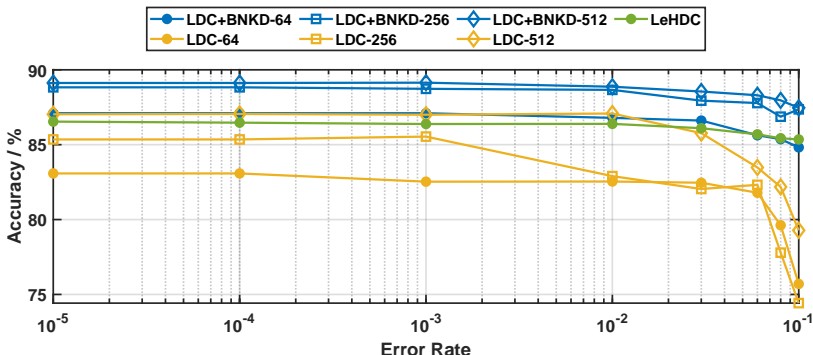

Figure 4: Bit-error robustness for selected VSA models, tested on FashionMNIST. We compare the LDC and LDC+BNKD models with the LeHDC model.

## F.6. Robustness Evaluation

The robustness of VSA models is of paramount importance. As vector dimension decreases, the models may become increasingly sensitive to bit errors, potentially impacting their performance. We investigate the bit error robustness in Figure 4. The results show that while high-dimensional VSA models indeed demonstrate excellent robustness against bit error, low-dimensional LDC models are less robust. However, our LDC+BNKD training method interestingly shows great robustness. Nevertheless, since hardware in practice will have a much lower bit-error rate than we tested, e.g., $< 10^{-6}$[43], even the vanilla LDC is still robust against normal bit errors [44].

## F.7. BN and KD on Binary Neural Networks

Table 16: The accuracy (%) of different models on FashionMNIST. The evaluation is conducted on their real model, binarized model, and the BN/KD assisted versions.

| | Real | Binary | Binary+BN | Binary+KD | Binary+BNKD |
|---|---|---|---|---|---|
| MLP | 90.29 | 87.20 | 88.45 | 88.39 | 89.12 |
| CNN | 90.47 | 83.67 | 84.59 | 85.53 | 86.14 |
| ResNet-18 | 92.15 | 91.31 | 91.96 | 91.93 | 92.25 |

We apply the BN and KD strategy to a 3-layer binary MLP, a shallow binary CNN (which has 3-layer convolution followed by a 2-layer classifier), and the binarized ResNet-18 on the FashionMNIST dataset, as a quick validation of interest. The results are presented in Table 16. We draw the same conclusion, as from the binary VSA training, that both BN and KD are beneficial for BNN optimization. However, there are also unique observations on each BNN. For example, the binarized CNN performs much worse than the real-valued counterpart, because we change the activation from ReLU to sgn(), which loses much information. On the other hand, we keep ReLU as activation in binary ResNet-18, and the trained BNN can almost achieve the same performance as its real-valued counterpart.

# G. Discussion

By generalizing our interpretation to other BNN networks, we evaluate the BN and KD assistance on BNN model in Appendix F.7, and address the following discussion:

**Importance of $b_{BN}$.** $b_{BN}$ shows little help during LDC training since the LDC model is centrosymmetric w.r.t. zero. However, for general BNN models, other asymmetric activations are involved, such as PReLU [15]. $b_{BN}$ will potentially play an important role in these cases, thus BN is still necessary For BNNs, and its lightweight expression as a threshold (Eq.9) is still preserved.

**Temperature in Knowledge Distillation.** For the KD setup, the choice of temperature $T$ is critical. Current works only evaluate the performance variation along different $T$ choices. However, the $T$ selection should be determined in advance based on the network architectures and classification tasks or adaptive during training. Since the large capacity difference between a well-performed teacher network and a BNN, it is also a potential effort to explore individual temperatures on each network. As an extension of our analysis, these could be the future effort on the KD for BNN training.

**Comparison and Relationship between LDC, VSA, and BNN.** Before LDC, VSA training is heuristic and involves random generation and iterative updates, resulting in high vector dimensionality. LDC proposes to train VSA in a BNN-like manner, optimizing binary vectors to significantly reduce dimensionality. (i) LDC is the first gradient-based method for VSA training, but its optimization remains basic and underexplored. This is the main motivation of our work to investigate the stable and adaptive dynamics in LDC training. (ii) Unlike BNNs, LDC is a partial-BNN architecture where the value mapping $\mathcal{V}(x)$ is real-valued and applied to individual values. This distinction makes LDC training different from typical BNNs, which often have a real-valued first layer accepting all features in input data and different activations. However, since the updates for **F** and **C** in LDC resemble BNN binary weight training, our analysis of LDC with BNKD can also be extended to BNN training.

