# OpenReview forum: "Towards Vector Optimization on Low-Dimensional Vector Symbolic Architecture"
_CPAL.cc/2025/Proceedings_Track — CPAL 2025 (Proceedings Track) Poster_

### Official Review · Reviewer_rnaX · 2025-01-10

**Rating:** 7
**Confidence:** 2

**Review:**

## Summary
This work explores the training dynamics of using low-dimensional computing (LDC) to train vector symbolic architectures (VSAs). In particular, the benefits of batch normalization and knowledge distillation are explored from both theoretical and empirical perspectives. The authors conclude that BN and KD are indispensable components to include when training VSAs using LDC.

## Strengths
* The paper is well written and provides a nice introduction to topics that have seen less direct attention from the ML research community of  late.
* VSAs are a promising potential direction to find highly efficient models for edge devices, but many open questions remain regarding their training.
* The proposed LDC+BN+KD empirical results in the best performance on a variety of small datasets.
* Figure and tables provide support for the authors claims.
* The proposed additions of BN and KD do not impose additional latency and only a minor impact on the memory footprint of the trained VSAs.

## Weaknesses
* No fundamental weaknesses were identified that would preclude acceptance. Some minor flaws/weaknesses listed below:
* Adding comparisons with other potential normalization strategies such as RMSNorm in addition to BN may be of some interest; however, this can be explored in future work. Authors may wish to include such a potential study direction in their future work discussions.
* Given the large capacity differences between the teacher and student networks in this setting, exploring the use of generalized Jensen-Shannon divergence may be a better approach than the typical KL divergence as the latter can result in mode seeking behaviour when used with a large capacity mismatches. See [1-3] for more info.
* Some of the language used is somewhat casual considering the content. I.e., “a potential game changer” (L27), “run fast to move away from zero” (L168). Generally, these do not detract from the paper but the authors may wish to use more precise language in some instances.
* Figure 3 -> Axis tick labels are very hard to read, suggest using a black font here.


[1] Ferenc Huszar. How (not) to train your generative model: Scheduled sampling, likelihood, adversary? ´ arXiv preprint arXiv:1511.05101, 2015.

[2] Tuan Anh Le. Reverse vs forward kl, December 2017. URL https://www.tuananhle.co. uk/notes/reverse-forward-kl.html.

[3] Agarwal, Rishabh, et al. "On-policy distillation of language models: Learning from self-generated mistakes." The Twelfth International Conference on Learning Representations. 2024.

---

### Official Review · Reviewer_17Dd · 2025-01-10

**Rating:** 5
**Confidence:** 3

**Review:**

### Overview

This paper explores optimization techniques for training binary vector symbolic architectures. It emphasizes the role of batch normalization and knowledge distillation in enhancing accuracy, while maintaining computational efficiency.

### Strengths

- The analysis of BN and KD's impact on gradient dynamics, optimization stability, and inference accuracy.
- The paper identifies and addresses critical gaps in binary VSA training, particularly in the stability of training dynamics.
- The paper is relatively clear.

### Weaknesses

- A detailed comparison of actual inference times is missing, leaving an important aspect of the method's practical efficiency unexplored.

- The ablation study of BN and KD require further investigation to fully understand their impact on the model's performance and efficiency.

---

### Official Review · Reviewer_6s3H · 2025-01-11
**Comprehensive analysis on a rather uncommon topic.**

**Rating:** 5
**Confidence:** 2

**Review:**

### Summary:
This paper explores Vector Symbolic Architecture (VSA) tailored for ultra-lightweight devices. In which the authors draw parallels between VSA and binary neural networks (BNNs) using the $\text{sgn}$ activation function and focus on a more efficient training paradigm termed Low-Dimensional Computing (LDC). This approach substitutes high-dimensional random vector generation with gradient-based optimization while utilizing real-valued counterparts of $\mathbf{F}$ and $\mathbf{C}$ and approximating $\text{sgn}(x)$ as $x$.
The authors provide a detailed analysis of training dynamics for feature vectors $\mathbf{F}$ and class vectors $\mathbf{C}$. Their key findings include the critical role of Batch Normalization (BN) in stabilizing feature vector learning and how Knowledge Distillation (KD) enhances prediction confidence and generalization.
The efficacy of their approach is demonstrated on various VSA tasks and real-world datasets, including FashionMNIST. The proposed LDC method, combined with BN and KD, surpasses state-of-the-art (SOTA) VSA training techniques in accuracy and model size.
### Strengths:
1) The authors present their work in a clear and systematic manner, making the VSA scheme easier to understand for readers without related background.
2) They offer a comprehensive analysis of LDC mechanisms, such as gradient distribution with/without BN and the role of BN in optimization.
### Weaknesses:
1) The practical relevance of VSA remains unclear, as the evaluations primarily focus on VSA tasks and toy datasets like FashionMNIST. It is also uncertain whether VSA has advantages over post-training quantization.
2) While LDC reduces the dimensionality D of $\mathbf{F}$ and $\mathbf{C}$, the approach seems to rely on gradient-based updates, resembling traditional randomized methods primarily; which seems to fall back to the field of BNN or network quantization.
### Questions:
Can you elaborate on why $\mathcal{V}(x)$ can be approximated by a shallow network? Could this be related to the low dimensionality of the data?

---

### Meta-Review · Area_Chair_umom · 2025-02-03

**Recommendation:** Accept (Poster)
**Confidence:** 3

**Metareview:**

This work focuses on improved training of low-dimensional computing (LDC) models designed for vector symbolic architectures (VSAs). The key findings are that batch normalization (BN) is critical for stabilizing feature vector learning in LDC classifiers, and knowledge distillation (KD) enhances prediction confidence and generalization.

Reviews were generally positive, though with some reservations. Multiple reviewers thought the paper was well-written and provides a nice introduction to LDC models on VSAs, and the open problems in training in such a framework. Some reviewers identified a few weaknesses in the experiments, asking for additional information and comparisons. For example, reviewer 17Dd requested inference times and an ablation study, and Reviewer rnaX requested comparisons with other normalization strategies such as RMSNorm among other changes. However, the authors have uploaded a revised version with additional experiments that address these concerns.

Reviewer 6s3H questioned the significance of the results, arguing that the practical relevance of VSAs remains unclear relative to existing techniques, such as post-training quantization. However, I think the authors adequately address this point in their rebuttal, which draws an analogy between the known benefits of quantization-aware training in binary neural networks over post-training quantization.

Overall, I think this work presents valuable insights for training LDC models on VSAs, and convincingly demonstrates the practical benefits of the proposed methods. While ML on VSAs may be considered a niche topic, I believe it is of sufficient interest to the CPAL community, and will find an audience at the conference, especially among researchers interested in ML on resource-limited devices. Therefore, my recommendation is an "accept".

---

### Decision · Program_Chairs · 2025-02-11

Accept (Poster)